# Mapping Soil Biodiversity in Europe and the Netherlands

**Michiel Rutgers** [1,*], **Jeroen P. van Leeuwen** [2] , **Dirk Vrebos** [3] , **Harm J. van Wijnen** [1],
**Ton Schouten** [1] **and Ron G. M. de Goede** [4]

1   National Institute for Public Health and the Environment (RIVM), Antonie van Leeuwenhoeklaan 9,
    3721 MA Bilthoven, The Netherlands; Harm.van.wijnen@rivm.nl (H.J.v.W.); ton.schouten@rivm.nl (T.S.)
2   Biometris, Wageningen University and Research (WUR), PO Box 16, 6700 AA Wageningen, The Netherlands;
    jeroen.p.van.leeuwen@gmail.com
3   Ecosystem management research group, Department of Biology, University of Antwerp,
    Universiteitsplein 1c, B2610 Antwerpen, Belgium; dirk.vrebos@uantwerpen.be
4   Soil Biology Group, Wageningen University & Research, PO Box 16, 6700 AA Wageningen, The Netherlands;
    ron.degoede@wur.nl
*   Correspondence: michiel.rutgers@rivm.nl

**Abstract:** Soil is fundamental for the functioning of terrestrial ecosystems, but our knowledge about soil organisms and the habitat they provide (shortly: Soil biodiversity) is poorly developed. For instance, the European Atlas of Soil Biodiversity and the Global Soil Biodiversity Atlas contain maps with rather coarse information on soil biodiversity. This paper presents a methodology to map soil biodiversity with limited data and models. Two issues were addressed. First, the lack of consensus to quantify the soil biodiversity function and second, the limited data to represent large areas. For the later issue, we applied a digital soil mapping (DSM) approach at the scale of the Netherlands and Europe. Data of five groups of soil organisms (earthworms, enchytraeids, micro-arthropods, nematodes, and micro-organisms) in the Netherlands were linked to soil habitat predictors (chemical soil attributes) in a regression analysis. High-resolution maps with soil characteristics were then used together with a model for the soil biodiversity function with equal weights for each group of organisms. To predict soil biodiversity at the scale of Europe, data for soil biological (earthworms and bacteria) and chemical (pH, soil organic matter, and nutrient content) attributes were used in a soil biodiversity model. Differential weights were assigned to the soil attributes after consulting a group of scientists. The issue of reducing uncertainty in soil biodiversity modelling and mapping by the use of data from biological soil attributes is discussed. Considering the importance of soil biodiversity to support the delivery of ecosystem services, the ability to create maps illustrating an aggregate measure of soil biodiversity is a key to future environmental policymaking, optimizing land use, and land management decision support taking into account the loss and gains on soil biodiversity.

**Keywords:** digital soil mapping; soil functions; soil biodiversity; ecosystem services; soil multi-functionality

## 1. Introduction

Healthy soil systems contain a very diverse assemblage of soil organisms, with a total biomass often exceeding that of aboveground organisms [1,2]. Soils vitally contribute to the functioning of ecosystems, for instance for providing a habitat for plants, animals and microorganisms, for water regulation, and for nutrient cycling. Moreover, soil biodiversity plays a key role in regulating the processes that underpin the delivery of many ecosystem services [3–8].

Despite the generally agreed importance of soil biodiversity for sustaining life on earth, there are not many data of soil biological attributes that represent extended soil gradients in time and in space [3,9,10] (with the exception of [11]). Hence, maps and trends exhibiting soil biodiversity are difficult to create, in contrast to maps of soil type, soil pH, and soil organic matter.

In 1992, soil biodiversity and its presumed decrease became a wider acknowledged issue with the ratification of the Convention on Biological Diversity (CBD). This brought wider attention to the beneficial use of biodiversity [12], besides the protection of species. In the EU thematic strategy for soil protection, a range of soil threats was defined, but without soil biodiversity loss [13]. The knowledge and the data on soil biodiversity were considered insufficient. Instead, research to underpin the relevance of soil biodiversity and the expected downward trends due to intensive land use and soil degradation was stimulated [14–17], including mapping and assessment approaches [18–21].

Recently, the 14th meeting of the Conference of the Parties (COP 14) of the Convention on Biological Diversity (CBD) held in Sharm El-Sheikh, Egypt, 17–29 November 2018, called for two important items: (1) To consider the preparation of a report on the state of knowledge on global soil biodiversity and make it available for consideration before COP15 (2020), and (2) planning of an international symposium in 2020 on global soil biodiversity.

The monitoring of soil biodiversity is hampered by a lack of clear definitions. Obviously, most soil organisms cannot be seen by the naked eye, yet the total biomass is immense. Soil microbes should be included in biodiversity assessment; yet microbial species richness is incredibly high. For soil functioning, the density of catabolic units is relevant, but this is not often part of a biodiversity analysis. Consequently, for a comprehensive view of soil biodiversity to assess the state of soils, we suggest a simple three-dimensional mental frame for collecting and analyzing soil data covering diversity (species, genes, biomolecules), activity (density, biomass, enzyme activities), and dynamics (food web interactions, networks, resilience and resistance; inspired after [22–26]). Ideally, data collection and analysis should address these dimensions independently.

Currently, there is no general consensus on how to set up biological soil monitoring [14]. Existing soil monitoring activities vary widely in their scope, goal, duration, efforts, and in the parts of the soil system that they represent. For specific research questions, monitoring of soil biological attributes has been undertaken, generally within a small environmental window, with only a few texture types, land uses, stressors, and types of organisms. We have no comprehensive overview but refer to descriptions in Turbé et al. (2010) [14] and Van Leeuwen et al. (2017) [9]. For our study, we used existing data and knowledge of the Netherlands soil monitoring network [11], and of a few selected EU projects [9,17].

Soil biological attributes that were feasible for routine measurements and that could be plausibly linked to soil biodiversity and life support functions were included in the Netherlands soil monitoring network (NSMN [11,27,28]), from the start in 1997 until the end of the last monitoring cycle in 2014. The selected attributes are soil community parameters of earthworms, nematodes, enchytraeids, and micro-arthropods and a large range of microbial parameters. Van Wijnen et al. (2012) [29] used these data for digital soil mapping of a specific soil function: The natural attenuation capacity of soils. The model was based on predictive environmental properties and selected biological soil data from the NSMN according to an expert model for this soil function.

Soil biodiversity maps at the continental level are more difficult to assemble, because of a lack of consistent data. A general state of soil biodiversity in Europe was first described in the European Atlas of Soil Biodiversity [30] with distribution maps of soil faunal groups (e.g., tardigrades, rotifers, nematodes, etc.), which show the estimated number of species per biogeographic area or country. The resolution in these maps is very low. In a follow up, the Global Atlas of Soil Biodiversity [31] presented global maps with predictions for soil fungi based on a digital soil mapping procedure and a map with a soil biodiversity index based on microbial biomass and 14 co-occurring soil fauna groups.

Recently, Aksoy et al. 2017 [32] produced a high-resolution map illustrating the soil biodiversity potential in Europe. The map was constructed from various spatially explicit layers with information of predictive properties for the quantification of soil biodiversity according to expert knowledge.

These properties were soil-pH, texture, soil organic matter, evapotranspiration, temperature, soil biomass productivity, and land use type. Aksoy et al. (2017) [32] compared their map with the maps of earthworm community metrics of Rutgers et al. (2016) [21] and found that these maps were statistically correlated.

In the EU FP7 project EcoFINDERS, for the first time, a European-wide schema was designed for monitoring soil biodiversity attributes and was tested in a pilot study with 81 locations [17,33]. The pilot contained samples from forestry, arable, and grassland systems in five climatic zones of Europe. Griffiths et al. (2016) [20] found a strong correlation between soil pH and bacterial diversity (TRFLP analysis) in these samples, and produced a bacterial diversity map of Europe. Earthworms were excluded from the monitoring for logistical reasons [33], but Rutgers et al. (2016) [21] instead collected and harmonized existing earthworm data from 3838 sites in several European countries and applied digital soil mapping in order to produce a European map with earthworm community parameters, i.e., abundance, richness, and Shannon–Wiener index.

Gardi et al. (2013) [15] produced a European map with threats to soil biodiversity as perceived by 20 experts, demonstrating the need for reliable information on the issue whether soil biodiversity decrease should be considered as a threat to soil quality affecting our natural environment.

The objective of this study is to demonstrate how data limitations in the first place, and limitations in the modelling of soil biodiversity in the second place, can be efficiently dealt with in a digital soil mapping approach. As proof of the two concepts, we produced two maps of the soil biodiversity and habitat function of soils, one for the Netherlands and one for Europe. This work in progress is a follow up of earlier publications on a regional scale (e.g., Scotland [34], France [35,36], Netherlands [29]), and European scale (e.g., [20,21,30,32,37]). We took advantage of the progress of the monitoring of soil biological attributes in the Netherlands [11] and in many EU-projects, i.e., SoilTrec [38], Soilservice [16], EcoFINDERS [17,33], and LANDMARK [9]. The operationalization of the soil biodiversity function as one of five overarching soil functions was performed according to the design in the EU H2020 project LANDMARK (Land management: Assessment, research, knowledge base [39,40]).

We used regression analyses to relate soil biological attributes to predictive environmental parameters for which high-resolution maps exist. We used models for quantification of soil biodiversity and habitat provisioning based on the set of soil attributes. With the models and the results from the regressions, we produced maps for the Netherlands and Europe. Because of limited data and the lack of standardized models for quantification of the soil biodiversity function, slightly different procedures had to be followed, all making efficient use of sparse information. Data from the NSMN were used for the map of the Netherlands. Data from the monitoring in the EU-project EcoFINDERS were used for the European map. Verification of the maps was done at two levels: Uncertain predictions from the steps in the DSM approach in the map of the Netherlands, and uncertain quantification of the soil biodiversity function in the map of Europe. The challenges and criticisms associated with mapping soil biological attributes and soil biodiversity are discussed.

## 2. Materials and Methods

### 2.1. Data Sources: Netherlands

Soil biological attributes were analyzed in the Netherlands soil monitoring network (NSMN) as described by Rutgers et al. (2009) [11]. The data from the first monitoring cycle (1999–2004) were compiled for multiple regression analysis as described in Van Wijnen et al. (2012) [29]. Generalized linear regression models (GLMs) of the Gaussian family were used to relate the soil biological attributes to predictive environmental properties [41]: Land use, soil type, and other environmental parameters (Table S1, Supplementary section). The models to be fitted with available data were all formulated to be according to the syntax (Equation (1)):

$$
\begin{aligned}
\text{Response biological attribute} = \\
\text{Intercept} + a \cdot \text{Loess} + b \cdot \text{Alluvial Clay} + c \cdot \text{Peat} + d \cdot \text{Sand} + e \cdot \text{Dairy grass} + \\
f \cdot \text{Arable} + g \cdot \text{Semi-natural grass} + h \cdot \text{Heather} + i \cdot \text{Mixed forest} + \\
j \cdot \text{Longitude} + k \cdot \text{Longitude}^2 + l \cdot \text{Latitude} + m \cdot \text{Latitude}^2 + n \cdot \text{SOM} + o \cdot \text{SOM}^2 + \\
p \cdot \text{pH} + q \cdot \text{pH}^2 + r \cdot \text{Clay-particles} + s \cdot \text{Clay-particles}^2 + t \cdot \text{Pal} + u \cdot \text{Pal}^2
\end{aligned}
\tag{1}
$$

where loess, alluvial clay, peat, sand (soil texture types), and dairy grass, arable, semi-natural grass, heather, mixed forest (land uses) are categories (integer: 0 or 1), SOM is soil organic matter (%), clay particles (<2 μm%), and Pal is phosphorus in ammonium lactate extraction (mg/L).

The quadratic terms for the scalar predictors in the regression formula allow for predicting non-linear responses as inflicted by optimum and minimum conditions.

The regression models were calibrated using a stepwise procedure based on the Bayesian information criterion (BIC) [42]. This is done in order to restrict the addition of terms to those that have a significant contribution to the overall model, making the full model highly significant. Calculations were conducted using S-Plus 2000 (MathSoft, Cambridge, MA). Subsequently, the regression formulae were used to generate high-resolution maps of soil biological attributes.

For almost all soil biological attributes in the NSMN, significant GLMs were obtained (Table S1; Supplementary Section), like: Bacterial activity (pmol leucine or thymidine/g dry soil/h), potential carbon mineralization rate (mg C/kg dry soil/wk), potential nitrogen mineralization rate (mg N/kg dry soil/wk), nematode density (n/100 g fresh soil), nematode richness (n taxa), lumbricidae density (n/m$^2$), lumbricidae richness (n taxa), enchytraeidae density (n/m$^2$), enchytraeidae richness (n taxa), micro-arthropod density (n/m$^2$), and micro-arthropod richness (n taxa). For more information on the monitoring design, the sampling, and analysis of soil biological attributes, see Rutgers et al. (2009, 2012) [11,28].

## 2.2. Data Sources: Europe

In 2012, the FP7 project EcoFINDERS organized a European sampling campaign with 81 sampling sites spanning three land uses (cropland, grassland, forestry), five climate zones (Atlantic, continental, Mediterranean, alpine, boreal), and gradients in texture, soil organic matter, and pH, as described by Stone et al. (2016) [33]. Besides potential environmental predictors, a range of soil biological parameters were analyzed: Nematode, enchytraeid, micro-arthropod, and protist community parameters, and many microbial community parameters (archaea, bacteria, fungi), some functional genes, and physiological profiles. The data were published in a set of articles in a special issue of Applied Soil Ecology in 2016 [17,20,21,33,43–47].

Regressions such as boosted regression trees and classical GLM did not produce suitable models for the predictive environmental properties (land use, climate zone, and other environmental and soil properties). For that reason, it was decided to use median values of the soil biological attributes in the categories of the sampling network (three land use types and five climatic zones) in the mapping procedure.

For logistical reasons, the sampling and analysis campaign in EcoFINDERS did not include the analysis of earthworms: Sampling takes a lot of time, and trained personnel should do the analysis. This was not feasible in the setup of the EcoFINDERS project. Instead, Rutgers et al. (2016) [21] collected earthworm data from existing data sources in several European countries. In total, 3838 records with earthworm data were collected in order to construct earthworm maps of Europe with digital soil mapping.

## 2.3. Modelling and Mapping Soil Biodiversity: Netherlands

The proxy-indicator system for modelling soil biodiversity and habitat provisioning in the Netherlands was based on eleven biological soil attributes, representing five groups of soil organisms: Microorganisms (C and N potential mineralization rates, leucine incorporation rate) and abundances and richness of communities of nematodes, enchytraeids, earthworms, and micro-arthropods. More information on the monitoring and the indicators can be found in Rutgers et al. (2009) [11]. For these

soil attributes, GLMs were derived from data of the first monitoring cycle (1999–2004) of the NSMN with predictive environmental properties (Table S1; Supplementary Section).

Maps with predictive environmental properties were the same as described in Van Wijnen et al. (2012) [29]. In every grid cell of the map, the value of the biological attribute was predicted with the regression models. The average value of all grid cells in the Netherlands was used in the calculation of the z-score, a standardized difference between the mean of a distribution and the value of an individual biological attribute [48]. In this way, the maps with biological attributes show relative values in standard deviations scaled to the national average value, which is set to zero. This procedure makes it possible to combine maps of very different soil biological attributes, into quantification of the relative soil biodiversity function.

In total, 11 maps with soil biological attributes were summed up using equal weights for each of the five groups of soil organisms (20% per group) and equal weights for the attributes within a group (with fauna: 50% for abundance and 50% for richness; with microbes: 33% for each attribute). In this way, quantification of soil biodiversity was obtained only with soil biological attributes, while a digital soil mapping approach was adopted based on predictive environmental properties, i.e., pH, SOM, texture, nutrients (N and P), land use, and coordinates [29]. This is in contrast to the European analysis (next paragraph), where information of chemical soil attributes was also used to model the soil biodiversity function.

### 2.4. Modelling and Mapping Soil Biodiversity: Europe

The proxy indicator system for modelling and mapping of the soil biodiversity function of Europe was based on 11 biological and chemical soil attributes: Total organic carbon (TOC), total nitrogen, total phosphorous, pH, clay content, three earthworm community metrics (abundance, richness, and Shannon index), microbial (bacteria plus fungi) biomass, bacterial biomass, and bacterial diversity. The process of attribute selection and ranking is described in Van Leeuwen et al. (2017) [9]. In short, expert knowledge was collected through questionnaires to gather scores for attributes using a logical sieve method [49]. Attributes were scored on relevance (R) and on sensitivity (S) towards four integrated attributes ('biology', 'nutrients', 'structure', and 'hydrology'). The integrated attributes aided in steering the usability of the information provided by the attributes towards the quantification of soil biodiversity. A weight factor for each of these integrated attributes towards soil biodiversity was also included in the questionnaire. The scores for relevance and sensitivity of the attributes for each integrated attribute and the weight factor of the importance of the integrated attributes were used to calculate a score for the attributes for soil biodiversity using the formula (Equation (2)):

$$X = \sum \frac{R + S}{2} W \qquad (2)$$

where $X$ = score for the attribute $X$, $R$ = relevance value of attribute $X$ to the integrated attributes, $S$ = sensitivity value of attribute $X$ to the integrated attributes, and $W$ = weighting factor of the importance of the integrated attributes (fraction; sum of weight factors is 1).

As the scores were, in some attributes, quantified at the aggregated level (organic C/N/P/K received a single score), these scores were disaggregated (fractions shown in Table 1). Besides a score for relevance and sensitivity, we also included an assessment on data availability and quality. For the predictive attributes based on maps of LUCAS top soil data [50], a score of 1 was used, because the maps were produced with a high density of point data across Europe. For the other attributes, linear regression models were used, based on data from the EcoFINDERS project (microbial biomass, bacterial biomass and diversity, and earthworm community [21,33]). For these attributes, the explained variation ($R^2$) of the model was used as measure for data quality. The scores for data quality were added to the logical sieve weights.

Some attributes (TOC, pH, clay content, and bacterial diversity) will not show a positive proportional relationship with the soil biodiversity function. For TOC, we used a ceiling of 15% to avoid overrepresentation of peat soils. For soil pH, an optimum was assumed at a pH($H_2O$) of

6.5 [36,51,52]. For clay content, an optimum was assumed at 25% clay particles, as lower values show a decrease in fertility and microbial biomass [53], and higher values may lead to anaerobic conditions [54]. For bacterial functional diversity, 1—hillslope from the community-level physiological profile determined in Biolog plates—was used as a measure for richness [47].

**Table 1.** Scores and final weight factors of six soil biological attributes and five chemical attributes in the model for soil biodiversity and habitat provisioning (see [9]).

| Attribute | Score Logical Sieve (X) | Fraction from Aggregated Score | Data Quality ($r^2$) | Final Weight Factor (W) | Adaptation of Dataset |
|---|---|---|---|---|---|
| %Total Organic C | 3.24 | 0.50 | 1 | 2.12 | if TOC < 15% Biodiv_TOC = TOC else Biodiv_TOC = 15% |
| Total N | 3.24 | 0.167 | 1 | 0.71 | |
| Total P | 3.24 | 0.167 | 1 | 0.71 | |
| pH | 3.14 | 1 | 1 | 4.14 | Biodiv_pH = (6.5 − abs(pH − 6.5)) |
| %Clay | 3.13 | 1 | 1 | 4.13 | Biodiv_Clay = (25% − abs(Clay − 25%)) |
| Earthworm abundance | 3.49 | 0.50 | 0.252 | 1.87 | |
| Earthworm Shannon | 3.49 | 0.26 | 0.267 | 0.97 | |
| Earthworm Richness | 3.49 | 0.24 | 0.249 | 0.90 | |
| Microbial biomass | 3.40 | 1 | 0.821 | 4.22 | |
| Bacterial abundance | 3.46 | 0.49 | 0.372 | 1.88 | |
| Bacterial diversity | 3.46 | 0.51 | 0.381 | 1.96 | Rich_Bacterial = 1.0 − Functional_Rich |

The attributes were plotted on the maps according to their z-scores taking the distribution of all data on the map into account. This resulted in maps with a total count for the z-scores of zero, i.e., at exactly the mean value for the attribute [48]. Each map was multiplied with the normalized weight factor from Table 1 and added to the integrated map. In this way, a proxy for the relative status of soil biodiversity and habitat provisioning was depicted on the map of Europe.

*2.5. Verification of Spatially Explicit Predictions: A Regional Study in the Netherlands*

The reliability of predictions of soil biological attributes on the map of the Netherlands was analyzed with data from the second monitoring cycle of the NSMN 2006–2011. Field observations in the NSMN from all sites in the province of Noord-Brabant (*n* = 49) were taken for a comparison with the values of soil characteristics on the maps and the soil biological attributes predicted by the regression models [55].

First, the actual soil type and land use of each site in the Province of Noord-Brabant was compared with the predicted soil type and land use. When there was a mismatch in the land use prediction, the nearest location with the correct land use was selected with the function NEAR in ArcGIS, and that was taken as the site for comparison.

Second, observations on soil biological attributes and soil characteristics at 49 sites in the second monitoring cycle of the NSMN were compared with the existing soil maps (described in [29]) and subsequently with the values for soil biological attributes predicted by the regression models (Table S1, supplementary section). From the 49 sites, all but one (discarded from the analysis) were on sandy soils (WSR: gleyic podzol): 29 dairy farms (grassland), 2 arable farms, 8 forest sites, 7 heather sites, and 2 semi-natural grasslands. Data from predictive environmental properties (SOM, pH, clay content, Pal) and predicted biological attributes (abundance and richness of earthworms, nematodes, enchytraeids, and micro-arthropods, and micro-organisms) were compared with the observed properties according to the equation (Equation (3)):

$$\text{Delta VAR}_{\text{pred|obs}} = (\text{VAR}_{\text{pred}} - \text{VAR}_{\text{obs}}) \cdot (\text{VAR}_{\text{obs mean}})^{-1} \tag{3}$$

where VAR is the environmental property or the biological soil attribute. The subscript 'obs' indicates the data from observations in the second monitoring cycle of the NSMN, and 'pred' indicates the input from existing soil maps [29] or from the regression analysis (this study).

*2.6. Verification of Predictions on Soil Biodiversity: Europe*

The soil biodiversity map of Europe in this study was correlated with the map produced by Aksoy et al. (2017) [32] in order to compare predictions of both models. For verification, Kendall's correlation coefficient was used. Also, a 'difference map' was produced, to visualize local deviations of both maps. For this, the European map was reclassified to 10 classes using a natural breaks algorithm available in ArcGIS. Break points were identified by choosing the class breaks that best group similar values and that maximize the differences between classes. This reclassified map was then subtracted from the map produced by Aksoy et al. (2017) [32], which also consists of 10 classes. Due to the low resolution of the map at European scale, a direct comparison with point data for verification was considered not to be realistic.

## 3. Results and Discussion

*3.1. Netherlands*

A relative measure for soil biodiversity and habitat provisioning (referred to as soil biodiversity) was calculated with the z-scores from 11 soil biodiversity attributes and plotted on the map of the Netherlands (Figure 1).

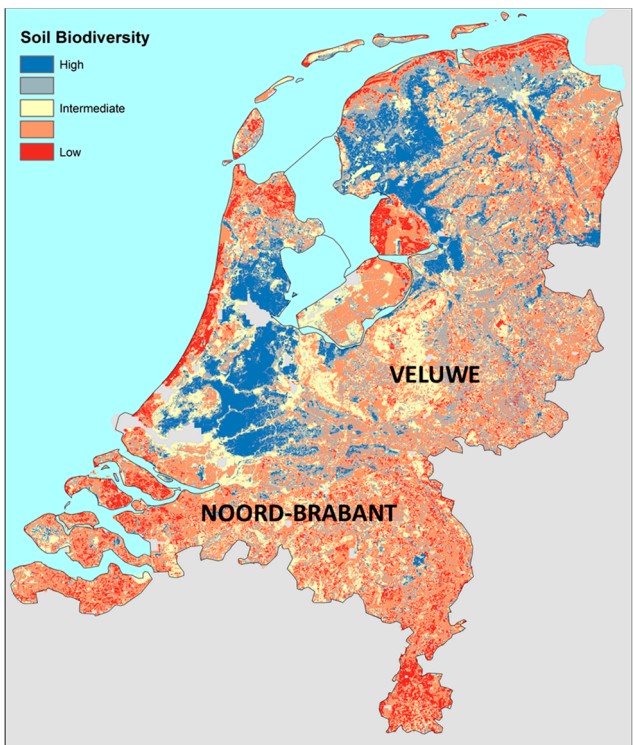

**Figure 1.** Predicted soil biodiversity in the Netherlands (relative; sum of z-scores). Predictions for 11 soil biological attributes were taken as a proxy for the calculation of the soil biodiversity: Abundances and richness of four soil fauna groups (earthworms, enchytraeids, nematodes, and micro-arthropods) and attributes for the microbial community (C and N mineralization rate and leucine incorporation rate). The data were obtained from the Netherlands soil monitoring network (NSMN; Rutgers et al. 2009).

Many gradients can be observed in this map, mostly agreeing with observations already described in the soil biodiversity literature, such as a lower soil biodiversity in arable (tillage) systems compared

to grassland systems, a lower soil biodiversity in sandy soils compared to loam and clay, and a high soil biodiversity in drained peat soils due to high bacterial activity and biomass [11,16]. It might be surprising to find an intermediate soil biodiversity in the 'Hoge Veluwe', a large nature area in the Netherlands (Figure 1, large yellow area in the center). One explanation might be the neglect of the fungal community as a building stone for the complete proxy indicator system. Fungi were included in the NSMN at a late stage [11], so not enough data were collected for regression analysis. It is known that the density and richness of fungal communities in this nature area is high (data from the second monitoring cycle of the NSMN).

Besides missing indicators for, e.g., the soil fungi, we did not derive specific weights for different soil attributes contributing to the quantification of soil biodiversity. Also, many modern biodiversity indices were not explored for applicability to quantify the soil biodiversity function. Consequently, this soil biodiversity map of the Netherlands should be seen as one step in a series to improve soil biodiversity mapping by more and reliable data and through underpinning with better biodiversity models, for instance such as the biodiversity model developed in the LANDMARK project [9,39].

### 3.2. Verification of Predictions of Soil Biological Attributes (Netherlands)

Confidence levels in mapping need to be addressed before we can make an exploration towards practical application. As a first step in addressing uncertainty in the maps presented here, the biodiversity map of the Netherlands was used to clarify the reliability of the predictions from the 'digital soil mapping' approach. Accordingly, the predictions for the condition of soil biological attributes were verified with data of 49 sites in the Province of Noord-Brabant, which were obtained independently in the second monitoring cycle of the NSMN (2006–2011). The difference in predicted and observed values for soil biological attributes can provide some insight to the error of the predictions, and confidence levels of the maps. The Province of Noord-Brabant was the first region in the Netherlands where this was tested [55].

The soil texture type in the maps was the same as found in the field, but this is not surprising, because the major soil texture type in the province of Noord-Brabant is sand (gleyic podzol: Approximately 90% of the total area). The land use was predicted poorly and was only correct for about 50% of the sites. This observation might come from a mismatch between the coordinates of the sampling and the coordinates of the farm registration in the database for administrative reasons, or from a change in the land use between the land use map (LGN6 [56]) and the land use during the sampling in the second round of the NSMN.

In Table 2 a summary of the comparison is presented. The observations for all soil biological attributes at 49 sites are on average 5% lower than the predictions from regression models; the average error is 5%. However, taking the errors in an absolute sense (positive and negative values are both adding to the error), the error is 45%. For agricultural grassland sites (29), the average error is very small (2%), but the absolute total error is comparable to that of forest and heather (Table 2). Semi-natural grasslands and arable systems were not analyzed separately because not enough data were available. The somewhat limited performance for forestry and heather is, besides the role of sample size, partly the result of relatively large overprediction of micro-arthropod abundance. On average, the predictions for micro-arthropod abundance were more than two times higher than observed.

Part of the error in modelled predictions of biological attributes also arises from differences in the values of the soil properties in the higher-resolution soil maps (e.g., soil organic matter, pH, nutrients (N and P), clay content [29]) with the actually measured soil properties in the NSMN 2006–2011. The error in map values and field data for soil parameters was, on average, 14% and the absolute total error was 30% (data not shown). The clay content was predicted poorly (−69%), but this was not an issue for concern, since sandy soils have a low clay content and no significant effect on the prediction is expected. Overall, the error in the chemical soil predictors also contributed to differences between predicted and observed condition of soil biological attributes, which is a disadvantage of the digital soil mapping approach [57].

**Table 2.** Difference between predicted and observed values (expressed in fractions) for 11 biological attributes in 49 soil samples of the Province Noord-Brabant [54]).

| | All Biological Attributes | Earthworm Richness | Earthworm Abundance | Enchytraeid Richness | Enchytraeid Abundance | Microarthropod Richness | Microarthropod Abundance | Nematode Richness | Nematode Abundance | Potential N-Mineralization | Potential C-Mineralization | Leucine Incorporation Rate |
|---|---|---|---|---|---|---|---|---|---|---|---|---|
| mean (49) | −0.05 | −0.08 | −0.23 | 0.01 | 0.00 | −0.19 | −0.69 | 0.10 | 0.02 | 0.03 | −0.01 | 0.45 |
| standard deviation | 0.57 | 0.33 | 0.56 | 0.32 | 0.81 | 0.42 | 1.34 | 0.20 | 0.64 | 0.39 | 0.57 | 0.73 |
| minimum | | −0.83 | −1.33 | −0.50 | −1.40 | −1.51 | −3.41 | −0.35 | −0.64 | −0.77 | −0.96 | −0.73 |
| maximum | | 0.52 | 1.04 | 1.14 | 3.51 | 0.51 | 1.81 | 0.41 | 2.19 | 0.63 | 1.32 | 2.26 |
| ABS mean | 0.45 | 0.27 | 0.48 | 0.24 | 0.56 | 0.35 | 1.02 | 0.19 | 0.44 | 0.33 | 0.47 | 0.63 |
| mean agr.grass (29) | 0.02 | −0.18 | −0.41 | 0.03 | −0.15 | −0.27 | 0.12 | 0.15 | 0.17 | 0.04 | 0.03 | 0.67 |
| ABS mean agr.grass | 0.43 | 0.30 | 0.54 | 0.23 | 0.69 | 0.29 | 0.43 | 0.17 | 0.48 | 0.33 | 0.47 | 0.82 |
| mean forest (8) | −0.35 | 0.16 | -0.04 | −0.08 | 0.16 | −0.50 | −2.16 | −0.23 | −0.57 | −0.20 | −0.59 | 0.14 |
| ABS mean forest | 0.54 | 0.16 | 0.15 | 0.16 | 0.43 | 0.63 | 2.16 | 0.25 | 0.57 | 0.31 | 0.59 | 0.52 |
| mean heath (7) | −0.14 | 0.09 | 0.08 | −0.19 | 0.19 | 0.18 | −2.15 | 0.11 | −0.03 | 0.18 | −0.03 | −0.01 |
| ABS mean heath | 0.41 | 0.09 | 0.08 | 0.29 | 0.39 | 0.26 | 2.22 | 0.11 | 0.22 | 0.40 | 0.35 | 0.08 |

'ABS mean' is absolute mean (difference) and indicates error. Between brackets, the number of sampling sites. The averages of semi-natural grasslands (2) and arable sites (2) are not shown separately.

*3.3. Europe*

The predicted soil biodiversity function for Europe is presented in Figure 2. Some observations in this map are confirmed with data on soil biodiversity in the scientific literature. For instance, the soil biodiversity is higher in grassland systems than in arable systems [11,16] the predicted low soil biodiversity in Lower Saxony and Denmark might be explained by the arable land use with abundant use of manure from the cattle industry. Furthermore, to the south, there seems to be a trend to a lower soil biodiversity coinciding with increased temperatures, lower humidity, and subsequently, a lower soil organic matter content. A higher soil biodiversity is demonstrated in the temperate and relatively wet areas in the west and north-west of Europe, especially in soils with high organic matter content.

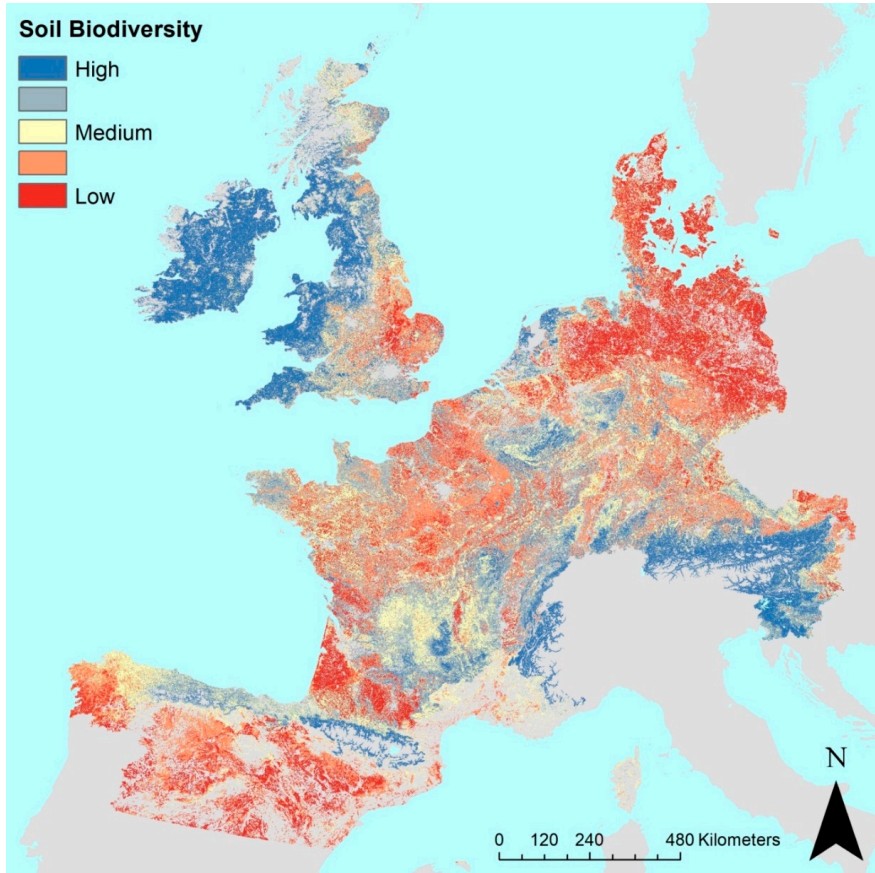

**Figure 2.** Predicted relative soil biodiversity in several European countries (sum of z-scores). Predictions for six biological soil attributes and five chemical soil attributes together were taken as a proxy for the soil biodiversity function, i.e., total organic C, total N, total P, pH, clay content, abundance, Shannon diversity and richness of earthworms, microbial biomass and bacterial abundance, and diversity. The data were obtained from the EU FP7 project EcoFINDERS and the LUCAS soil survey [17,33,58].

*3.4. Comparison of Models for Assessing Soil Biodiversity (Europe)*

In order to investigate the performance of our model and available data for assessing soil biodiversity on the European map as request to validate our map, we used the model and map of Aksoy et al. (2017) [32]. True validation of a soil biodiversity map is impossible as outlined in the introduction. To test whether the maps tend to agree or disagree, we did a correlation analysis. Per raster cell the correlation between the maps appeared to be very significant (Kendalls' tau = 0.41, $p < 0.0001$; Figure 3). Consequently, there is no reason to discard any map based on this correlation. Of course, there is a chance that both maps are 'overall' wrong, but this chance becomes smaller when more data and knowledge contributing to the conceptual soil biodiversity model become available.

For a more thorough analysis, we subtracted the map in Figure 2 from the map presented by Aksoy et al. (2017) [32] (Figure 4). A striking difference between the soil biodiversity maps is visible in 'Les Landes' (south-west of France) where we predict a relative low soil biodiversity while Aksoy et al. (2017) [32] predicted an intermediate soil biodiversity. On the other hand, in Brittany (very west part of France), our predictions show an intermediate soil biodiversity, while the predictions by Aksoy and colleagues are low. The cause of mismatches might be also found in the combined effects of a limited amount of data, a lack of representativeness (i.e., problems associated with our ignorance to quantify soil biodiversity), errors in the predictive soil properties, and spatial and temporal variation. Also, differences in the expert judgement-based conceptual models for quantification of soil biodiversity in both maps will be illuminated in the difference map (Figure 4).

Although both maps predict soil biodiversity differently, the fact that the maps are correlated means that there is no need to examine the origin of the differences. Furthermore, as long as we do not have a key to improving the models for the mapping and assessment of soil biodiversity and we are still lacking data for verifying sources of uncertainty, finalization of a soil biodiversity model is impossible. Consequently, our conclusion is that both maps add to the knowledge base about quantification and assessment of soil biodiversity.

The model for quantifying soil biodiversity of Aksoy and colleagues differs from our model, in the sense that we included soil biological attributes. They labeled their map as a 'potential' soil biodiversity map, acknowledging that only 'indirect' indicators (chemical, physical, and land use parameters) for soil biodiversity were used. The inclusion of biological attributes would possibly improve the maps as we have shown here, because of the use of more direct information. However, we acknowledge that available soil biological data is generally insufficient for mapping and that 'indirect' soil attributes are a practical alternative. Although we advocate the use of biological attributes for quantification of the soil biodiversity function, we are aware that, also with this information, there is still no consensus on how to quantify and assess soil biodiversity in the very complex soil system [3,25,26].

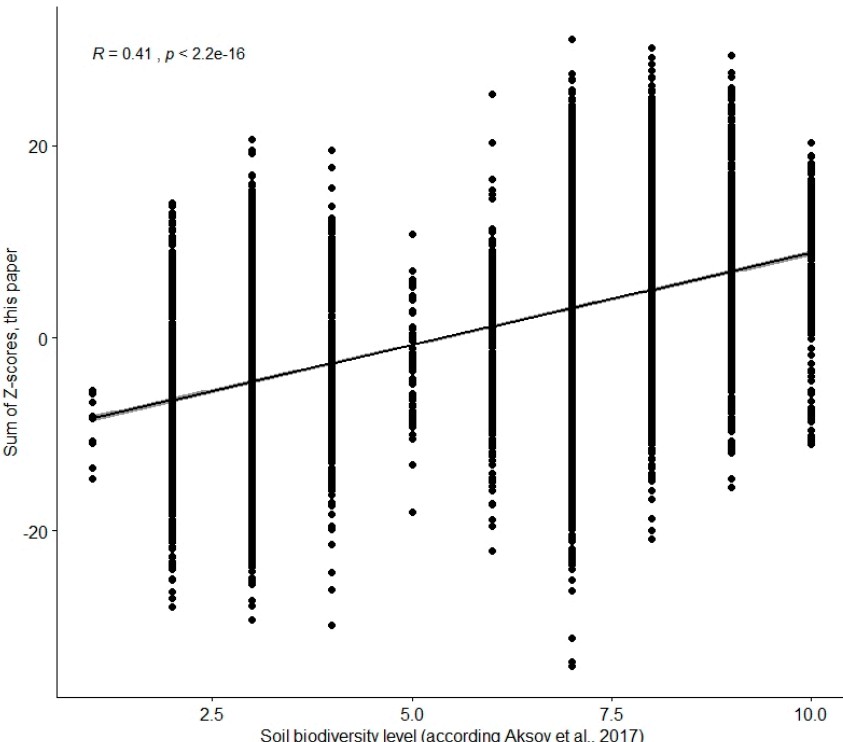

**Figure 3.** Result of a comparison of data of the soil biodiversity map of Aksoy et al. (2017) [32] with the data of the map in Figure 2. The relative status of the soil biodiversity was quantified by Aksoy and colleagues in 10 discrete levels. The quantification of the soil biodiversity with z-scores in this paper, was continuous.

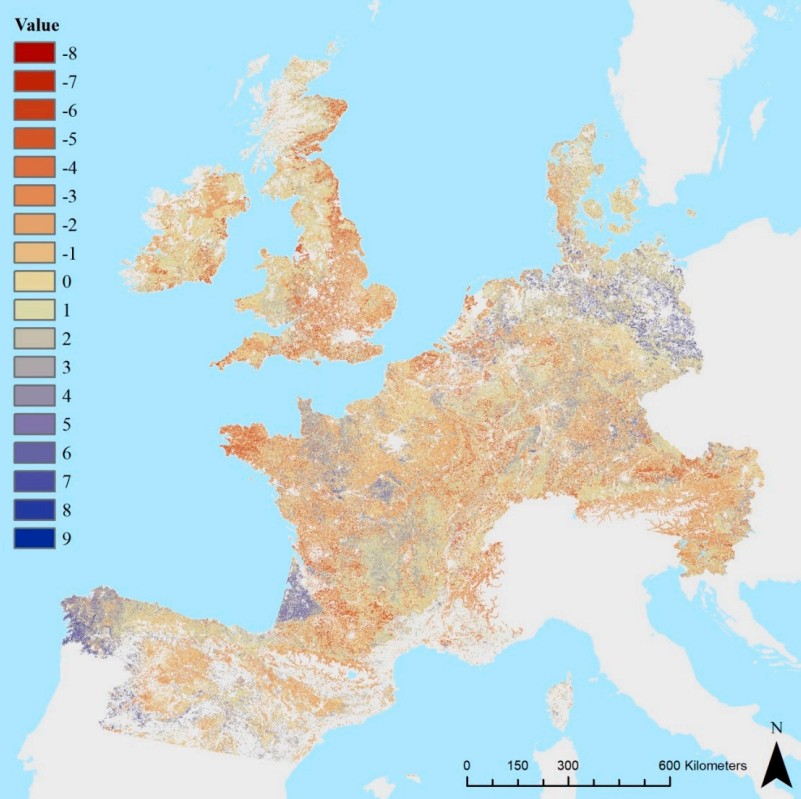

**Figure 4.** Map showing the difference between the soil biodiversity map of Aksoy et al. (2017) [32] and the map in Figure 2. Positive numbers signify a higher estimation by Aksoy and colleagues.

For practical soil monitoring, we would like to stress that there is no reason to not include biological attributes, despite the perception that collecting such data produces highly variable results and is costly to conduct. In the Netherlands soil monitoring network, we have proven that it is feasible to include affordable, sensitive, reliable, and robust soil biological indicators in a practical soil-monitoring network [11], and to display this information on a map ([29,55] this study). Consequently, it seems a good moment to start collecting data and monitor soil biological attributes in Europe and worldwide, as some researchers did already in several European pilots with soil biological attributes: In EcoFINDERS [33], LANDMARK [9], and in the monitoring schema for LUCAS 2018 (https://esdac.jrc.ec.europa.eu/projects/lucas).

In the European soil biodiversity model, we considered both biological and abiotic soil attributes as useful, using all available information as efficiently as possible, despite the sometimes-unfortunate low signal-to-noise ratio of some attributes. We combined the information and applied a multi-criterion 'logical sieve' weighting approach for including relevant attributes in the final model, irrespective of the type of the attribute [9,49].

It must be stressed that the data from the 81 sites in the FP7 EcoFINDERS project had severe limitations: Too few samples for too many categories and gradients, i.e., three land use types, five climate zones, and gradients in soil texture, soil organic matter, and pH. The regressions were not significant, except for the relationship between pH and microbial diversity [20]. For the earthworm data, the regressions with 3838 sites were significant but only had an explained variation of about 25% due to, e.g., different sampling protocols, taxonomic identification errors, and missing sampling dates [21]. In the Dutch soil monitoring with one sampling protocol, the regressions for the earthworm community had higher explained variations (52% and 61% for abundance and richness, respectively; Table S1, Supplementary Section). Despite limitations, the data from the EcoFINDERS project covered a representative part of Europe and better data spanning such a large area were not available. Sampling

more sites and harmonization of protocols for sampling and analysis will quickly result in better regressions and thus better maps.

*3.5. Reducing Uncertainty in Modelling and Mapping*

Any map will display values for illustrative parameters that are 'uncertain' to some extent. We have shown that the current state of mapping and assessment of soil biodiversity is uncertain in two aspects and needs two types of verification. First, the spatial resolution in information from real data is often (much) lower than the spatial resolution we expect the map to provide. Second, large uncertainty arises from the 'endpoint problem', since there is no consensus on how to quantify soil biodiversity [26,59,60].

Digital soil mapping (DSM) approaches are popular to accommodate the problem of spatial data with a low resolution [57]. We applied DSM to obtain spatial information with a higher resolution for the Netherlands and verified the accuracy of the individual attribute maps with independent data of biological and chemical soil observations. For the Province of Noord-Brabant in the Netherlands, it seems that the observations for soil biodiversity attributes were, on average, somewhat lower than was predicted from the regression analysis of the Netherlands, but the differences were not big. In our opinion, the total difference is limited, and becomes almost insignificant when the 11 attributes used in this study are combined in one proxy indicator system for soil biodiversity. As a general rule, the increments to improve any multidimensional model for soil biodiversity with not-yet included attributes, will decrease when the number of included attributes is increased. From the combination of multiple attributes in one proxy-indicator system, we conclude that predictions of soil biological attributes in the Netherlands with the DSM approach are feasible and useful for a first step to assess the soil biodiversity function in a spatial context.

Currently, there is not yet consensus on how to quantify and assess the soil biodiversity function in a unified approach [26,59,60]. However, we adopted a proxy-indicator for soil biodiversity, designed to maximally utilize limited data and information. The notion we used is that any attribute that can be statistically or mechanistically linked to the soil biodiversity function can be used for quantification, i.e., not missing one of the three dimensions in the mental framework for soil biodiversity quantification. In many soil protection acts, such an approach is already the basis for deriving environmental quality criteria for ecological risk assessment of soil contamination with species sensitivity distributions (SSD [61]). When more knowledge on assessing soil biodiversity becomes available, structured conceptual soil biodiversity models and differential weights might be put in place [9,62].

Although already a reasonable number of relevant attributes was collected in the NSMN, showing convergence to quantification of the soil biodiversity function, we are aware that there is always a chance that potentially critical attributes were not part of the models. For example, the NSMN did not contain indicators for monitoring of, e.g., protists, while fungi were only sampled in the second monitoring cycle. Bacterial diversity information was also not used for the biodiversity map of the Netherlands.

Despite the overwhelming complexity of the living soil system and the challenge to select the 'right' attributes, we think it is possible to design appropriate, practical, and affordable monitoring systems for soil biodiversity and habitat provisioning, once our data and knowledge base is developed to maturity with data from real systems over large areas. Such a reference view on soil biodiversity and soil biological attributes is appearing now in the Netherlands. Consequently, dedicated proxy-indicator systems can be designed for specific land uses, soil types, climate zones, and environmental stresses, which are smaller than the Dutch monitoring system.

## 4. Conclusions and Prospective

In this study, the feasibility of using soil biological data for assessing and mapping the soil biodiversity function was demonstrated. Although the model we applied for quantification of soil biodiversity is conceptually in development, and the realized predictions of soil biological attributes in

the digital soil mapping approach are statistically optimized, we hypothesize that the maps contribute to the developments in this area, and we hope to add some value to the worldwide assessment of soil biodiversity by 2020 as requested by the UN-CBD (COP 14). Improvement of data availability together with further steps of conceptualization are areas that jointly need attention for constituting to a knowledge and policy base for soil biodiversity.

　　Soil biodiversity and habitat provisioning is only one of the soil functions. The EU working group on mapping and assessing of ecosystems and their services (MAES) is preparing guidelines for supporting EU member states to reach the targets in the EU biodiversity strategy, i.e., action 5: To map and assess the state of ecosystems and their services in the national territories [11]. Recently, the EU commissioned the MAES working group to perform a soil pilot study for preparing guidelines to assess and map the multi-functionality of soils, i.e., contributions to a bundle of soil ecosystem services [18]. In the H2020 project, LANDMARK tools for assessing and mapping of five overarching soil functions are developed [39,40]: (1) Soil biodiversity and habitat provisioning (this paper), (2) primary productivity, (3) water regulation and purification, (4) climate regulation, and (5) nutrient cycling and regulation. In other studies, sometimes more functions are acknowledged, but often they are addressed in one index ('soil health'). As a final remark, multiple soil functions should be represented in the soil monitoring and assessment for addressing the total natural capital of soils, for instance in LUCAS.

**Supplementary Materials:** The following Tables are available online at http://www.mdpi.com/2571-8789/3/2/39/s1: Table S1. General Linearized regression Models (GLMs) of data on selected soil biological attributes in the first monitoring cycle in the Netherlands Soil Monitoring Network [11]; Table S2. Biological attributes contributing to the soil biodiversity model for the soil biodiversity map of Europe. Data were obtained from the EU FP7 project EcoFINDERS [17,21,33].

**Author Contributions:** Conceptualization, M.R., J.P.v.L., T.S., and R.G.M.d.G.; data curation, J.P.v.L., H.J.v.W., and T.S.; investigation, J.P.v.L., D.V., H.J.v.W., and T.S.; methodology, M.R. and T.S.; validation, M.R. and D.V.; writing—original draft, M.R. and J.P.v.L.; writing—review and editing, T.S. and R.G.M.d.G.

**Funding:** This research did not receive any grant from funding agencies in the private, commercial and not-for-profit sector.

**Acknowledgments:** We thank Ece Aksoy for scientific support and the data of the potential soil biodiversity map of Europe. We obtained data from the EU-FP7 project EcoFINDERS (grant number 264465). The work was partly done within the EU H2020 project LANDMARK (grant number 635201). We thank LANDMARK consortium members for their valuable contributions to the discussions on the modelling and mapping of soil functions. The work was also partly done within the RIVM projects with the Biological Indicator for Soil Quality (BiSQ) and the Netherlands Soil Monitoring Network (NSMN). The RIVM projects M/607604, M/607406 and E/607063 were commissioned by the Ministry of Infrastructure and Water Management and the Province of Noord-Brabant.

**Conflicts of Interest:** The authors declare no conflict of interest.

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
