# Peer review of "Mapping Soil Biodiversity in Europe and the Netherlands"

_soilsystems, doi:10.3390/soilsystems3020039_

Round 1
Reviewer 1 Report
Please find attached a marked-up pdf with comments and most relate to english expression and some minor points of clarification. Figure 2 is also NOT included as part of the manuscript. Also the smaller locality in the Netherlands needs to be highlighted where it is and proper soil classifications used. Please make sure the pdf is referred to and where blue marks there is alternate text.

Author Response
Dear reviewer 1:
Thank you for thorough looking at our manuscript. We adopted all remarks (see also PDF), which resulted in improved texts. All your remarks were very useful. Thank you. We added the reference of Lobry de Bruyn because it is indeed demonstrating the point we want to make.
I upload the PDF with my responses point by point. I had some difficulties to keep track in this PDF. Alternatively, I can make a long list of all points.
Sincerely
Michiel Rutgers

Reviewer 2 Report
The manuscript soilsystems-496016, titled „Mapping soil biodiversity in Europe and the Netherlands” by Rutgers et al. presents an approach for mapping soil biodiversity with limited data and models. The topic is interesting and timely, as well illustrated by the authors. So far, only few approaches to map soil biodiversity exist, and some of these are restricted to countries or specific groups of soil biota. These other approaches are mentioned in the manuscripts introduction, and I do not have anything to add to this (but I am sure I do not know all existing mapping approaches). The novelty here is the approach chosen by the authors, which is also compared to the soil biodiversity maps by Aksoy et al. 2017.
I am overall supportive of the manuscript as I believe such maps to be highly important in order to give voice to the importance and protection of soil biodiversity (also in the context of soil functions). To me, the study design and methods seem to be appropriate (if my understanding is correct), although I had some problems of understanding. Thus, I have the following critical remarks and recommendations which I hope the authors will find useful.
1) It is not completely clear why two different analyses were used for the Netherlands and for Europe. There are some differences between the models (e.g. no climate zones in the Netherlands model). The Netherlands model was validated against data from the same region, but from another sampling campaign, while the European model is compared to the model by Aksoy et al. 2017. These differences should be stressed. Furthermore, the question arises why the Netherlands model was not used and up-scaled to Europe (i.e. use the Netherlands regressions to predict European biodiversity), but a different modelling approach was chosen. It would be interesting to see whether that works, but I understand that the Netherlands model does not contain some parameters which are needed at the European level (e.g. climate zones).
2) Sections 2.1 – 2.4: I propose to change the structuring of these sections. The two data source sections already partially contain the regressions (i.e. modelling). I propose to write 1 section for the data sources (combined for the Netherlands and Europe) and to move the regressions to the respective modelling and mapping sections. Some lack of clarity which is caused by the current section structuring could be eliminated by that, e.g. on line 135 it is not given what these soil biological attributes are (but on line 183 and following).
3) Modelling: Some details about the modelling/regressions are not clear.
a) in equation 1, it is not clear what is meant by Loess, peat, clay-particles etc. Is this simply a yes/no question, i.e. variable “a” only remains in the equation when the soil is Loess (1), but is eliminated when it is not Loess (0)? This needs to be clarified.
b) Line 170 following: Were the methods the same as described above for the Netherlands model?
c) Line 182 following: I do not fully understand all the steps taken here (maybe an explanatory graph on the two different approaches would help). What I think was done (shortly summarized): Data from NSMN were used for GLMs to connect the soil biological attributes to the predictive environmental properties. These models were used to produce maps. These maps were summed to produce one soil biodiversity map. Now, what is the last sentence (l200-202) supposed to tell me? Soil biodiversity was not modelled with soil biological attributes alone, but based on the predictive environmental properties.
d) Netherlands verification: How and why was the soil type predicted? I do not understand why soil type, clay etc were not taken from maps as they belong to the parameters to predict biodiversity. The value of the regressions is that based on available maps biodiversity can be predicted.
Minor comments:
L 46 [9, 10] but [11] -> this sounds strange. Maybe change to “but see [11]”?
L 72 In my eyes, Figure 1 is not needed as it does not have much informational value. Instead, I propose to add the explanation from the figure legend to the main text, and also move parts of the paragraph lines 451 – 463 to the introduction, as both the figure legend and the discussion paragraph better explain the mental framework.
L 133/L187: the dates for the first monitoring cycle differ (1999 – 2003 vs. 1999 – 2004)
L 136: and other environmental parameters. Please state which parameters or refer to the model/Table S1...?
L 150-152: I’m not sure I understand this sentence right. What is meant by continuous soil maps regression formulae?
L 153: significant GLMs were obtained (Table S1): How do we know that a GLM is significant by looking at Table S1? Are only the remaining (significant) attributes shown here, or is the significance indicated somehow?
L 154 Bacterial biomass is mentioned in the text as significant, but missing in Table S1.
L 219 X is the score for the attribute X, according to the explanations for R and S
L 237 (Table 1): Maybe add to the column captions the abbreviations used in eq. 2 (e.g. final weight factor = W; Score logical sieve = X...). The column “Adaptation of dataset” is a bit cryptic in the expressions, maybe choose another way of presentation here. Biodiv_Clay: is 40 right here, or should that be 25 (as in the text)? Rich_Bacterial is not explained in the text.
L 247 Noord-Brabant (49) -> what does 49 stand for? The number of sites?
L 271: “our map and...” could be shortened to “both maps”
L 282 Figure 2 is missing, so I cannot evaluate the results
L 326 The low number of sites could also play a role (compared to agr.grass with 29 sites)
L 340 (Table 2): What do the colors mean? (orange and purple for microarthropod abundance)?
L 344 following: This partially belongs to the methods/is redundant
L 363 This is more a comparison with another model than a verification
L 371: Why was the comparison done? What are the differences? This is stated on lines 387-388, but the reader already wonders...
L 372: This sounds as if the Aksoy map is figure 5, but this is the result from subtracting one map from the other
L 376 – 378: What is meant by representativeness, which kind of errors in the predictive soil properties (wrong measurements? Wrong model prediction?)?
L 379 both teams -> which teams? Maybe better: both models?
L 430 As an outlook, can we infer that sampling more sites (81 in EcoFINDERS vs. 3838 in the Netherlands) will result in better regressions and thus better maps?
L 446-447 How do you know that additional, not-yet included attributes will improve the map only slightly, if they are not included? Does this refer to the attributes excluded from the regression models by selection based on BIC?
L 502: Is there no funding by LANDMARK to mention?
L 687 Table S1 What are the values presented in the table? The regression’s slopes? What do the colors stand for?
Typing errors (not all-encompassing):
L 87 space missing before [30]
L 90 space missing before [31]
L 98 wrong reference style (Rutgers et al. (2016a) -> [xx])
L 100 European wide schema -> Europe-wide (not sure here)
L 101 in a pilot -> in a pilot study?
L 115 parenthesis missing after [37] -> [37])
L 119 parenthesis missing after [40]
L 141 Longitide² -> Longitude²
L 147 T missing (The regression models)
L 186 parenthesis missing after [11]
L 197 summed together -> summed up?
L 228 parenthesis missing after [33]
L 236 circumstances -> maybe better: conditions?
L 244 section number missing
L 253/272 ARCGIS or ArcGIS?
L 260 P-Al is named Pal before (L144)
L 267 change section number (see L 244)
L 304 wrong section number
L 331 parenthesis missing after [29]
L 339 delete “2012b).”
L 340 (Table 2): Microarthropod richness and Leucine incorporation rate have capital letters, all other columns do not; Row ABSolute mean -> change to ABS mean?
L 351 form -> from
L 351 there seems a trend -> there seems to be a trend
L 363 wrong section number
L 380: space missing before “5”
L 383 i missing in mapping
L 384 we still lack data/ we are still lacking data
L 431 wrong section number
L 434 need -> needs
L 459 of missing (one of the three dimensions)
L 489: there is a , instead of a . after multi-functionality
L 688 Table is called Table S1 in the paper
P(Al)² is Pal² before
Bactyreial -> Bacterial
Author Response
Dear Reviewer 2
Thank you for this very timely review, which considerably improved our manuscript. I copied your comments in this repsons, followed by our arguments and approval (indicated with #), plus reference to the new lines in the improved manuscript (prepared for a resubmittal; for convenience I upload this version in the reviewers corner).
The manuscript soilsystems-496016, titled „Mapping soil biodiversity in Europe and the Netherlands” by Rutgers et al. presents an approach for mapping soil biodiversity with limited data and models. The topic is interesting and timely, as well illustrated by the authors. So far, only few approaches to map soil biodiversity exist, and some of these are restricted to countries or specific groups of soil biota. These other approaches are mentioned in the manuscripts introduction, and I do not have anything to add to this (but I am sure I do not know all existing mapping approaches). The novelty here is the approach chosen by the authors, which is also compared to the soil biodiversity maps by Aksoy et al. 2017.
I am overall supportive of the manuscript as I believe such maps to be highly important in order to give voice to the importance and protection of soil biodiversity (also in the context of soil functions). To me, the study design and methods seem to be appropriate (if my understanding is correct), although I had some problems of understanding. Thus, I have the following critical remarks and recommendations which I hope the authors will find useful.
1) It is not completely clear why two different analyses were used for the Netherlands and for Europe. There are some differences between the models (e.g. no climate zones in the Netherlands model). The Netherlands model was validated against data from the same region, but from another sampling campaign, while the European model is compared to the model by Aksoy et al. 2017. These differences should be stressed. Furthermore, the question arises why the Netherlands model was not used and up-scaled to Europe (i.e. use the Netherlands regressions to predict European biodiversity), but a different modelling approach was chosen. It would be interesting to see whether that works, but I understand that the Netherlands model does not contain some parameters which are needed at the European level (e.g. climate zones).
#1 Answer:
Thank you for your timely review and your support for this type of results. We agree that there is a need for such maps, as a communication tool and to enhance awareness on this particular type of biodiversity. This question is indeed relevant and we have tried to better explain why we have chosen this set up (see below). It is not ideal we admit and we see it as ‘work in progress’. There were two constraints; limited data and un-validated biodiversity models. We cannot solve this, and instead we illustrate how to deal with this. Both constraints are adding to a substantial amount of uncertainty in soil biodiversity assessments. From the survey point of view the soil biological data in the Netherlands are superb compared to many other databases, but unfortunately the monitoring has been stopped (in 2015). In the meanwhile modelling of soil biodiversity is improving e.g. thanks to the many EU projects addressing soil functioning. Given these positions, it was feasible to illustrate the issue of uncertainty through variation with the Dutch data and demonstrate the issue of conceptual uncertainty (ignorance) with the European models (one of Aksoy et al, and one of LANDMARK).
Added L 114-117: The objective of this study is to demonstrate how data limitations in the first place, and limitations in the modelling of soil biodiversity in the second place, can be efficiently dealt with in a digital soil mapping approach. As a prove of the two concepts we produced two maps of the soil biodiversity and habitat function of the soils, one of the Netherlands and one for Europe.
2) Sections 2.1 – 2.4: I propose to change the structuring of these sections. The two data source sections already partially contain the regressions (i.e. modelling). I propose to write 1 section for the data sources (combined for the Netherlands and Europe) and to move the regressions to the respective modelling and mapping sections. Some lack of clarity which is caused by the current section structuring could be eliminated by that, e.g. on line 135 it is not given what these soil biological attributes are (but on line 183 and following).
#2 Answer: We consider regressions not as biodiversity modelling, in order to separate the steps required for the DSM approach with that of quantification of the soil biodiversity function. DSM has a clear goal, i.e. trying to predict values of attributes for a set of conditions from existing data on measured attributes at several conditions. It is preferred to use those predictions only to interpolate (inside the Netherlands). We consider the Dutch regressions of very limited value to extrapolate outside the Netherlands (indeed the climate gradient in the Netherlands is too small for that). For that reason, we have chosen to separate regressions of attributes from modelling of soil biodiversity. The regressions are described in the Mat and Met section (it is merely a technical issue), and biodiversity modelling (a science discipline in development) in the results section.
3) Modelling: Some details about the modelling/regressions are not clear.
a) in equation 1, it is not clear what is meant by Loess, peat, clay-particles etc. Is this simply a yes/no question, i.e. variable “a” only remains in the equation when the soil is Loess (1), but is eliminated when it is not Loess (0)? This needs to be clarified.
#3 Answer: we clarified the input data for these attributes. In L 153-155: Where loess, alluvial clay, peat (soil texture types), and dairy, arable, semi-natural grass, heather, mixed forest (land uses) are categories (integer; 0 or 1), SOM is soil organic matter (%), clay particles (< 2 µm %), and Pal is ammonium lactate extraction of P (mg/l)
b) Line 170 following: Were the methods the same as described above for the Netherlands model?
#4 Answer: We used the same and more sophisticated regression techniques for the EcoFinders data. We assume that the much smaller dataset and many more environmental gradients in Europe are the cause for a lack of significant patterns in the data as stated in:
L 180-184. Regressions such as Boosted Regression Trees and classical GLM did not produce suitable models for the predictive environmental properties (land use, climate zone and other environmental and soil properties).
c) Line 182 following: I do not fully understand all the steps taken here (maybe an explanatory graph on the two different approaches would help). What I think was done (shortly summarized): Data from NSMN were used for GLMs to connect the soil biological attributes to the predictive environmental properties. These models were used to produce maps. These maps were summed to produce one soil biodiversity map. Now, what is the last sentence (l200-202) supposed to tell me? Soil biodiversity was not modelled with soil biological attributes alone, but based on the predictive environmental properties.
#5 Answer: The regressions were part of the DSM approach (see also #2) and are needed to extrapolate measured attributes (1300 sites) to map the Netherlands (numerous grid cells). Indeed, combinations of many maps made a picture of the soil biodiversity function (we call it ‘modelling’), for the Netherlands this was only a set of soil biological attribute maps (11 attributes). We adopted this semantics, in order to make differences in the approach of the Netherlands and that of Europe more clear, where also maps of chemical soil attributes were used to ‘model’ the soil biodiversity function.
L125-130 reformulated sentences:
We used regression analyses to relate soil biological attributes to predictive environmental parameters for which high-resolution maps exist. We used models for quantification of soil biodiversity and habitat provisioning based on the set of soil attributes. With the models and the results from the regressions we produced maps of the Netherlands and Europe. Because of limited data and the lack of standardized models for quantification of the soil biodiversity function, slightly different procedures had to be followed, all making efficient use of sparse information.
The reason is that in NL we had sufficient data to only use biological information. For Europe, we do not have sufficient soil biological information, and so we decided to also use chemical soil attributes (note: Aksoy et al. only used non-soil biological data for modelling soil biodiversity).
We think the issue is also made more clear now in L 213-215:
This is in contrast to the quantification of soil biodiversity of Europe (next paragraph), where also information of chemical soil attributes was used.
d) Netherlands verification: How and why was the soil type predicted? I do not understand why soil type, clay etc were not taken from maps as they belong to the parameters to predict biodiversity. The value of the regressions is that based on available maps biodiversity can be predicted.
#6 answer
See also the previous answer. Soil type and clay etc were taken from the maps to predict the value of a soil biological attribute. A specific stack of 11 maps with soil biological attributes was considered as a model for relative soil biodiversity (z-score calculation, plus expert-based weight factors).
Minor comments:
#7 All Done, see subnumbers below
L 46 [9, 10] but [11] -> this sounds strange. Maybe change to “but see [11]”?
#7.1 Done
L 72 In my eyes, Figure 1 is not needed as it does not have much informational value. Instead, I propose to add the explanation from the figure legend to the main text, and also move parts of the paragraph lines 451 – 463 to the introduction, as both the figure legend and the discussion paragraph better explain the mental framework.
#7.2 Done. deleted Fig 1. Moved parts to introduction
L 133/L187: the dates for the first monitoring cycle differ (1999 – 2003 vs. 1999 – 2004)
#7.2 Corrected
L 136: and other environmental parameters. Please state which parameters or refer to the model/Table S1...?
#7.3 Done
L 150-152: I’m not sure I understand this sentence right. What is meant by continuous soil maps regression formulae?
#7.4 Changed sentence to L162: Subsequently, the regression formulae were used to generate high-resolution maps of soil biological attributes.
L 153: significant GLMs were obtained (Table S1): How do we know that a GLM is significant by looking at Table S1? Are only the remaining (significant) attributes shown here, or is the significance indicated somehow?
#7.5 Indeed, only most significant parameters are coloured. We indicated this in the legend
L 154 Bacterial biomass is mentioned in the text as significant, but missing in Table S1.
#7.6 Corrected, by deleting biomass
L 219 X is the score for the attribute X, according to the explanations for R and S
#7.7 Added
L 237 (Table 1): Maybe add to the column captions the abbreviations used in eq. 2 (e.g. final weight factor = W; Score logical sieve = X...). The column “Adaptation of dataset” is a bit cryptic in the expressions, maybe choose another way of presentation here. Biodiv_Clay: is 40 right here, or should that be 25 (as in the text)? Rich_Bacterial is not explained in the text.
#7.8 We added a sentence on this: For bacterial functional diversity, 1 – hillslope from the community-level physiological profile determined in Biolog plates was used as a measure for richness [47].
L 247 Noord-Brabant (49) -> what does 49 stand for? The number of sites?
#7.9 Changed to N = 49
L 271: “our map and...” could be shortened to “both maps”
#7.10 Done
L 282 Figure 2 is missing, so I cannot evaluate the results
#7.11 Inserted the Fig (renumberd to 1)
L 326 The low number of sites could also play a role (compared to agr.grass with 29 sites)
#7.12 Added this as possible explanation
L 340 (Table 2): What do the colors mean? (orange and purple for microarthropod abundance)?
#7.13 Removed Colours
L 344 following: This partially belongs to the methods/is redundant
#7.14 We deleted that part
L 363 This is more a comparison with another model than a verification
#7.15 We renamed the title L383 -> comparison
L 371: Why was the comparison done? What are the differences? This is stated on lines 387-388, but the reader already wonders...
#7.16 We reformulated the first sentences:
In order to investigate the performance of our model and available data for assessing soil biodiversity on the European map as request to validate our map, we used of the model and map of Aksoy et al. [32]. True validation of a soil biodiversity map is impossible as outlined in the introduction.
L 372: This sounds as if the Aksoy map is figure 5, but this is the result from subtracting one map from the other
#7.17 Figs are renumbered, and we checked the right references.
L 376 – 378: What is meant by representativeness, which kind of errors in the predictive soil properties (wrong measurements? Wrong model prediction?)?
#7.18 Added: (i.e. problems associated to our ignorance to quantify soil biodiversity)
L 379 both teams -> which teams? Maybe better: both models?
#7.19 Done
L 430 As an outlook, can we infer that sampling more sites (81 in EcoFINDERS vs. 3838 in the Netherlands) will result in better regressions and thus better maps?
#7.20 We added: Sampling more sites and harmonization of protocols for sampling and analysis will quickly result in better regressions and thus better maps.
L 446-447 How do you know that additional, not-yet included attributes will improve the map only slightly, if they are not included? Does this refer to the attributes excluded from the regression models by selection based on BIC?
#7.21 sentence changed to (L467):
As a general rule, the increments to improve any multidimensional model for soil biodiversity which not-yet included attributes, will decrease when the number of included attributes is increased.
L 502: Is there no funding by LANDMARK to mention?
# 7.22 The primary funder of LANDMARK is at governmental level (EU), like the Dutch government for the RIVM project
L 687 Table S1 What are the values presented in the table? The regression’s slopes? What do the colours stand for?
#7.23 We added explanation of colours: The predictive environmental properties marked yellow and orange had respectively a positive and negative significant contribution to the model.
Typing errors (not all-encompassing):
# 8 All adopted
L 87 space missing before [30]
L 90 space missing before [31]
L 98 wrong reference style (Rutgers et al. (2016a) -> [xx])
L 100 European wide schema -> Europe-wide (not sure here)
L 101 in a pilot -> in a pilot study?
L 115 parenthesis missing after [37] -> [37])
L 119 parenthesis missing after [40]
L 141 Longitide² -> Longitude²
L 147 T missing (The regression models)
L 186 parenthesis missing after [11]
L 197 summed together -> summed up?
L 228 parenthesis missing after [33]
L 236 circumstances -> maybe better: conditions?
L 244 section number missing
L 253/272 ARCGIS or ArcGIS?
L 260 P-Al is named Pal before (L144)
L 267 change section number (see L 244)
L 304 wrong section number
L 331 parenthesis missing after [29]
L 339 delete “2012b).”
L 340 (Table 2): Microarthropod richness and Leucine incorporation rate have capital letters, all other columns do not; Row ABSolute mean -> change to ABS mean?
L 351 form -> from
L 351 there seems a trend -> there seems to be a trend
L 363 wrong section number
L 380: space missing before “5”
L 383 i missing in mapping
L 384 we still lack data/ we are still lacking data
L 431 wrong section number
L 434 need -> needs
L 459 of missing (one of the three dimensions)
L 489: there is a, instead of a . after multi-functionality
L 688 Table is called Table S1 in the paper
P(Al)² is Pal² before
Bactyreial -> Bacterial

Reviewer 3 Report
In this paper, the authors present the results of mapping soil biodiversity for five different groups of organisms at the scale of the Netherlands and Europe, on the basis of several soil habitat characteristics, that were not necessarily overlapping between the two maps. For the Netherlands map, generalized linear models were used to relate biodiversity metrics to habitat characteristics. For the European map a logical sieve method was used to select habitat factors and weightings for the different biodiversity attributes. For the Netherlands map, prediction verification with data from Noord-Brabant indicated that biological attributes were assigned with values 5% higher, on average compared to the actual observations. For the European map, differences between an existing soil map were obtained by subtraction. The map generated in this study differed from the Aksoy map markedly for some regions, although the maps were 41% correlated to each other. The authors conclude that there is a shortage of curated biodiversity data and a lack of consensus in categorizing biodiversity indicators to be used in soil mapping currently, in a routine and unified matter. Indirect indicators that are more commonly collected during soil monitoring/mapping efforts might be more useful in the interim.
I appreciate the authors' transparency about gross discrepancies in their mapping results compared to the other data products used for uncertainty-related analyses in the paper. Mapping biodiversity is critical and a continuously evolving effort in understanding anthropogenic land impacts. I think that this study is a useful contribution to the literature because you are demonstrating different avenues how data types can be assembled and weights assigned to map these biological attributes. I think there are several things that could make the paper clearer to readers.
INTRODUCTION: I think that the information articulated in lines 48-71 could be condensed, emphasizing the COP recognition of the need to monitor soil biodiversity and the paucity of comprehensive biodiversity data for mapping.
Maybe a comment on the more commonly mapped types of biodiversity data (in relation to the five types of organisms you use in your analyses) and availability of maps for these data (relating to the Netherlands and/or Europe) would be useful to mention so that readers know where the bulk of data collection/curation has been focused historically. Would it be possible to generate a concise table that summarizes the extent (i.e. presence of regular monitoring efforts/monitoring data and geographic breadth) of biodiversity data collection for microorganisms, nematodes, lumbricidae, enchytraeidae, and microarthropods for either Netherlands or Europe?
The objective of your study needs to be more clearly stated. I feel like this is a "methods demonstration" paper, or something along these lines, so clarifying that you are presenting two approaches to building soil biodiversity maps would be helpful in framing this paper as an exploratory or guidance document.
MATERIALS AND METHODS: Lines 138-143 - please make the Model equation text more clearly separated from the body text
TABLE 1: Is the bacterial richness and functional richness note meant to be a correction factor, as this attribute is not described in your methods text, like the adjustments/ceilings for pH, clay, and TOC
RESULTS AND DISCUSSION: Figure 2 appears to be missing from the draft
For possible inclusion in your discussion: Do you think that the foci for biodiversity monitoring should be contextualized around land-use type, so that maps can be compared on the basis of land use, which might reduce the burden of broad biodiversity monitoring efforts (i.e. subsets of different organisms might have different relevance depending on the lens of commodification, inherent biological richness, etc)?
Author Response
Dear reviewer,
Thank you very much form your timely review. It was very helpful to improve our manuscript. To see what we have done, I copied your comments below, followed by our answer and approval (indicated by #). We make references to improved sections and lines in the manuscript. For convenience I attach the revised manuscript to this review corner (also to show you the 'lost' Figure of the Netherlands).
Reviewer 3:
In this paper, the authors present the results of mapping soil biodiversity for five different groups of organisms at the scale of the Netherlands and Europe, on the basis of several soil habitat characteristics, that were not necessarily overlapping between the two maps. For the Netherlands map, generalized linear models were used to relate biodiversity metrics to habitat characteristics. For the European map a logical sieve method was used to select habitat factors and weightings for the different biodiversity attributes. For the Netherlands map, prediction verification with data from Noord-Brabant indicated that biological attributes were assigned with values 5% higher, on average compared to the actual observations. For the European map, differences between an existing soil map were obtained by subtraction. The map generated in this study differed from the Aksoy map markedly for some regions, although the maps were 41% correlated to each other. The authors conclude that there is a shortage of curated biodiversity data and a lack of consensus in categorizing biodiversity indicators to be used in soil mapping currently, in a routine and unified matter. Indirect indicators that are more commonly collected during soil monitoring/mapping efforts might be more useful in the interim.
I appreciate the authors' transparency about gross discrepancies in their mapping results compared to the other data products used for uncertainty-related analyses in the paper. Mapping biodiversity is critical and a continuously evolving effort in understanding anthropogenic land impacts. I think that this study is a useful contribution to the literature because you are demonstrating different avenues how data types can be assembled and weights assigned to map these biological attributes. I think there are several things that could make the paper clearer to readers.
#9 Answer: We thank this reviewer for the support to publish this, despite the ‘unconfidence’ we should have in biodiversity maps in general, and the maps in this study in particular.
INTRODUCTION: I think that the information articulated in lines 48-71 could be condensed, emphasizing the COP recognition of the need to monitor soil biodiversity and the paucity of comprehensive biodiversity data for mapping.
#10 Answer: We shortened this part with 3 lines:
This text was removed: Later on, this was evolved in terms of natural capital and the delivery of ecosystem services, e.g. in the Millennium Ecosystem Assessment [12]……As a result of the efforts for constructing a future EU Soil Framework Directive the focus on…….but without taking into account without the threat of a presumed loss of soil biodiversity loss.
Maybe a comment on the more commonly mapped types of biodiversity data (in relation to the five types of organisms you use in your analyses) and availability of maps for these data (relating to the Netherlands and/or Europe) would be useful to mention so that readers know where the bulk of data collection/curation has been focused historically. Would it be possible to generate a concise table that summarizes the extent (i.e. presence of regular monitoring efforts/monitoring data and geographic breadth) of biodiversity data collection for microorganisms, nematodes, lumbricidae, enchytraeidae, and microarthropods for either Netherlands or Europe?
#11 Answer
To be honest, I find this a bit tricky, because there are many rather small datasets with specific soil organisms and set ups in order to answer a specific research question, and almost no databases and set ups which grasp more land uses, more stresses and multiple groups of organisms. We can name a few, but we are sure that we miss data, and in that sense the added value is limited. To my knowledge, the breath of soil biological data is quite homogeneously spread over Europe (the larger ones), but local flavor determines the type of the organisms studied (such as earthworms and microbes in France, microarthropods in Italy, nematodes in the Netherlands, etc). I suggest to refer to existing literature reviewing soil biological indicators and data, such as from Turbe et al. 2010 and Van Leeuwen et al. 2017. Therefore, we made the following adjustments to the text:
In L 71 we added the following paragraph:
Currently, there is no general consensus on how to set up biological soil monitoring [9, 14]. Existing soil monitoring activities vary widely in their scope, goal, duration, efforts and in the parts of the soil system that they represent. For specific research questions, monitoring of soil biological attributes has been undertaken, generally within a small environmental window, covering only a few texture types, land uses, stressors, and types of organisms. We have no comprehensive overview of soil monitoring studies, but refer to descriptions in Turbé et al. (2010) [14] and Van Leeuwen et al. (2017) [9] For this study we used existing data and knowledge of the Netherlands Soil monitoring network [11], and of a selection of EU projects.
The objective of your study needs to be more clearly stated. I feel like this is a "methods demonstration" paper, or something along these lines, so clarifying that you are presenting two approaches to building soil biodiversity maps would be helpful in framing this paper as an exploratory or guidance document.
#12 Answer
This was also comment 2 of reviewer 1. We inserted the following sentences (and deleted one):
Deleted: In this study we present two maps with predictions for the soil biodiversity and habitat function in Europe and in the Netherlands
Added L114: The objective of this study is to demonstrate how data limitations in the first place and limitations in the modeling of soil biodiversity in the second place can be efficiently and transparently dealt with in a digital soil mapping approach. As a prove of the two concepts we produced two maps of the soil biodiversity and habitat function of the soils, one of the Netherlands and one of Europe.
MATERIALS AND METHODS: Lines 138-143 - please make the Model equation text more clearly separated from the body text
#13 Answer
Done
TABLE 1: Is the bacterial richness and functional richness note meant to be a correction factor, as this attribute is not described in your methods text, like the adjustments/ceilings for pH, clay, and TOC
#14 answer
We corrected this by adding the following L248:
For bacterial functional diversity, 1 – hillslope from the community-level physiological profile determined in Biolog plates was used as a measure for richness [47].
RESULTS AND DISCUSSION: Figure 2 appears to be missing from the draft
#15 answer
We are sorry for this inconvenience and corrected it in the current version of the manuscript.
For possible inclusion in your discussion: Do you think that the foci for biodiversity monitoring should be contextualized around land-use type, so that maps can be compared on the basis of land use, which might reduce the burden of broad biodiversity monitoring efforts (i.e. subsets of different organisms might have different relevance depending on the lens of commodification, inherent biological richness, etc)?
#16 answer
Yes, we think that can be envisaged, although also other stratification methods should be considered (climate zone, soil type). And that is also not a panacea, since it is not the analysis what makes it expensive, but the field visits, the handling of the samples, administration, data management, etc. In the Dutch monitoring system for example, these costs were substantial. To our surprise, the monitoring of earthworms was amongst the most expensive indicators (big samples, delicate and time consuming processing needed).
We added the following paragraph in the discussion
L489: Despite the overwhelming complexity of the living soil system and the problem to select the ‘right’ attributes, we think it is possible to design appropriate, practical and affordable monitoring systems for soil biodiversity and habitat provisioning, once our data and knowledge base is developed to maturity with data from real systems over large areas. Such a reference view on soil biodiversity and soil biological attributes is appearing now in the Netherlands. Consequently dedicated proxy-indicator systems can be designed for specific land uses,

Round 2
Reviewer 2 Report
The revised manuscript soilsystems-496016, titled „Mapping soil biodiversity in Europe and the Netherlands” by Rutgers et al. has been substantially improved. The manuscript is written much clearer now and has been convincingly revised. I am also satisfied with the author’s answers on why some issues were not addressed.
I have the following minor suggestions for further improvement:
1) Line 332 and line 351: The locations mentioned here (Hoge Veluwe and Noord-Brabant) are most likely only known to Dutch readers. Maybe these areas could be marked in the figure? This is not necessary, but would be nice to have.
2) The in-text citation style is inconsistent, e.g. Aksoy et al. 2017 [32] (line 107), Rutgers et al. (2016) [21] (line 112), Rutgers et al. [21] (line 120)
3) Line 498-500: Something is wrong in this sentence. Maybe it should be “with” instead of “which”?
4) The text is written in italics following equations 1 and 3
5) L346: confidence levels need (instead of needs)
6) L385: de -> is the
7) L 430: mean -> means
8) L 514: one “of” the three dimensions?
9) L536: delete the “be”
Author Response
Dear Reviewer,
We took all your comments, including the positions on the map of the Netherlands, except the format change from italics to normal text immediately after the equations. This text provides information about the equations. However. if the style of this journal does not allow for this, we will change it to normal text as you have indicated.
The changes were somewhat shifted compared to your indications. For your convenience, we made small changes in L470 (3), L325 (5), L363 (6), L403 (7), L477 (8), L500 (9). Between brackets the link to the number of your comment. I do not know whether the figures have the required resolution, but that will be arranged during the production.
Thank you for your review and quick comments.
Kindest regards
Michiel Rutgers
